# New Generation Gepants: Migraine Acute and Preventive Medications

**DOI:** 10.3390/jcm11061656

**Published:** 2022-03-16

**Authors:** David Moreno-Ajona, María Dolores Villar-Martínez, Peter J. Goadsby

**Affiliations:** 1Basic and Clinical Neurosciences, Institute of Psychiatry, Psychology and Neuroscience, King’s College London, London SE5 9PJ, UK; david.1.moreno_ajona@kcl.ac.uk (D.M.-A.); maria.villar_martinez@kcl.ac.uk (M.D.V.-M.); 2NIHR-Wellcome Trust King’s Clinical Research Facility/SLaM Biomedical Research Centre, King’s College Hospital, London SE5 9RS, UK; 3Department of Neurology, University of California, Los Angeles, CA 90095, USA

**Keywords:** gepants, CGRP, acute medications, painkillers, migraine

## Abstract

Migraine is a debilitating disease whose clinical and social impact is out of debate. Tolerability issues, interactions, contraindications, and inefficacy of the available medications make new options necessary. The calcitonin-gene-related peptide (CGRP) pathway has shown its importance in migraine pathophysiology and specific medications targeting this have become available. The first-generation CGRP receptor antagonists or gepants, have undergone clinical trials but their development was stopped because of hepatotoxicity. The new generation of gepants, however, are efficacious, safe, and well tolerated as per recent clinical trials. This led to the FDA-approval of rimegepant, ubrogepant, and atogepant. The clinical trials of the available gepants and some of the newer CGRP-antagonists are reviewed in this article.

## 1. Introduction

The relevance of headache and migraine in particular has been underestimated for many years, although we are now more aware of the disability these may lead to. Indeed, migraine is, after back pain, the most disabling condition worldwide [1]. Even during the COVID-19 pandemic, the relevance of headache remains. In fact, the SARS-CoV2 virus has been related to headache [2] and headache seems to be a prominent feature of the so-called long COVID [3]. Several treatments have been tested for the latter unsuccessfully [4]. Regarding migraine, medical treatment can be divided into acute and preventive medications [5]. Despite many patients benefitting from both, there are still patients where insufficient efficacy, tolerability issues, or contraindications prevent them from receiving a good treatment for their migraine attacks [6,7]. In general, most migraine treatments to date are based on drugs that were not specifically designed to treat migraine [8]. That has changed as our understanding of migraine pathophysiology has evolved and several neurotransmitters have shown a different degree of implication in headache generation [9]. Of these, the calcitonin-gene related peptide (CGRP) is the paradigm of bench-to-bedside medicine. From the first preclinical studies that showed its potential role in a cat model [10] to the studies in humans which confirmed its capacity to trigger migraine attacks [11]. The following step, with the development of drugs targeting this pathway, started with the first generation of CGRP receptor antagonists: the gepants [12]. Despite their utility, most compounds were not developed further due to hepatic side effects for formulation or commercial considerations. First generation gepants were recently covered by us elsewhere [13]. Apart from the monoclonal antibodies targeting the CGRP pathway [14], new gepants have shown positive results in terms of efficacy, tolerability, and safety [13]. Here, we aim to review the latest evidence involving the next generation gepants.

## 2. Material and Methods

We completed a narrative literature review search on the PubMed and Cochrane databases in March 2021, using the keywords: “CGRP small receptor antagonists”, “CGRP antagonists”, “gepants”, “next generation gepants”, “CGRP acute medication”, and “gepants migraine prevention”. Phase I to III randomized placebo-controlled clinical trials were included. First generation gepants, namely, olcegepant, telcagepant, BI44370TA, and MK-3207 were not included in the review. Publications in English and Spanish were included in our search. In total, 26 published studies were reviewed. Additionally, to understand the evidence behind these findings, previous studies where the role of CGRP in migraine was investigated were also included. All the articles were peer-reviewed publications from January 2000 to March 2021. Additional data was included from abstracts recently presented at the American Headache Society, the American Academy of Neurology, and the International Headache Society meetings. When no other source was available, press releases focusing on the latest trials were included.

## 3. Results

The results discussed below are summarized in Table 1.

### 3.1. Rimegepant

Rimegepant is the only drug of its class which has shown positive results both as a migraine acute and preventive medication. It was developed at the Bristol Myers Squibb Laboratory (New York, NY, USA) as an oral drug. Its absolute bioavailability has been reported up to 64% [25]. Following intake, the orally disintegrating tablet (ODT) formulation reaches a maximum concentration at 1½ h [15]. The volume of distribution is 120 L and plasma protein binding is very high, up to 96%. Blood-brain barrier penetrance seems irrelevant as oral doses of up to 300 mg/Kg led to a ratio of 0.19 at 24 h post-dose utilizing the Sprague—Dawley rat. CYP3A4 hepatic metabolism has been reported and, indeed, rimegepant exposure increases with the concomitant intake of CYP3A4 inhibitors, such as itraconazole [17,26]. Elimination half-life has been documented at 11 h with excretion being fundamentally through faeces. Regarding pharmacodynamics, the study in cynomolgus monkeys showed an IC_50_ of 0.14 ± 0.01 nM with the pKi being 10.6. The Ki in humans has been reported at 32.9 pM. Binding affinity and specificity to the CGRP receptor was documented with the only remarkable exception of the amylin 1 receptor (AMY1) showing an IC_50_ of 2.28 nM, as opposed to 54.3 pM for CGRP [26,27,28]. Recent studies have further investigated the possibility that rimegepant may also block the AMY1 receptor [29]. Indeed, an experiment in transfected Cos7 cells that expressed CGRP and AMY1 receptors demonstrated that rimegepant can effectively antagonize both [29]. The importance of amylin in migraine has been suggested [30,31], but is still ill-understood [5]. A total of three randomized clinical trials (RCTs) have been published for the acute treatment of migraine, with one recent RCT published showing positive results of rimegepant as a migraine preventive [23,32,33].

#### 3.1.1. Acute Treatment Studies

In 2014, using an adaptive dose design [34], the initial dose-defining RCT which also included sumatriptan 100 mg as active comparator included 799 participants who were treated following a single attack [35]. The 75-mg, 150-mg, and 300-mg doses were efficacious as compared to placebo. Rimegepant was well tolerated with only minor side effects [35]. The positive results of this initial phase II study opened the doors to the following phase III RCTs: studies 301 (NCT03235479) [16] and 302 (NCT03237845) [17] which were similarly designed and obtained analogue results. The former involved 1485 participants, whereas the latter was performed on 1186 participants who treated a single migraine attack with either 75 mg of rimegepant or the placebo. As for all relevant headache studies since the International Headache Society’s (IHS) recommendation in 2012 [36], the primary efficacy endpoint was pain freedom at 2 h. This was obtained by 19.6% of patients on rimegepant, while this figure was 12% for those on placebo; absolute difference: 7.6% (95%CI 3.3 to 11.9; *p* < 0.001). In line with US regulatory requirements, the co-primary outcome of freedom of the most bothersome symptom (MBS) was utilized, these being either nausea, phonophobia, or photophobia. Freedom from the MBS at 2 h for those who took rimegepant 75 mg was 37.6% in contrast with 25.2% with placebo (*p* < 0.001) [17]. Side effects were again mild, with nausea being the most common side effect reported. Similar to its predecessors, study 303 (NCT03461757) was a multicentre, double-blind RCT in which 732 patients treated a single migraine attack with rimegepant 75 mg ODT and 734 patients had a placebo [18]. As for studies 301 and 302, primary efficacy endpoints were positive. Pain freedom at 2 h with rimegepant ODT occurred in 21% of patients as compared to 11% on placebo (*p* < 0.0001). The ODT formulation was capable of achieving statistically significant pain relief in a higher proportion than the placebo as soon as 1 h after intake (36.8% vs. 31.2%; *p* < 0.05). Sustained response was also measured and was superior in the active group (13.5%) than in the placebo group (5.4%) when checked at 48 h. In line with this, 86% of the patients on rimegepant did not use rescue medication, these were NSAIDs or acetaminophen. Again, there were no safety issues and, in contrast with the first generation gepants [13], no hepatotoxicity was found. Since February 2020, rimegepant 75 mg ODT has been approved for the treatment of acute migraine by the United States Food and Drug Administration (FDA) [37].

#### 3.1.2. Preventive Treatment Study

The efficacy of rimegepant 75 mg as a migraine preventive medication (NCT03732638) has been documented in a recent publication [23]. In total, 747 patients were randomised to receive either rimegepant 75 mg or a placebo OD every other day over a period of 12 weeks after they had completed a baseline 4-week observation period using an electronic headache diary. The primary efficacy endpoint was a change in the mean number of migraine days per month in the last month as compared to the baseline. Of note, the baseline frequency was 10.3 (±3.2) in the active group and 9.9 (±3) in the placebo group. Between week 9 and 12 there was a reduction of 4.3 migraine days (95% −4.8 to −3.9) with rimegepant, while with the placebo there was a reduction of 3.5 days (difference: −0.8; 95% −1.5 to −0.2; *p* = 0.099). Interestingly, the reduction in mean monthly migraine days was already significant at the end of week 4 (−2.9, 95%CI −3.3 to −2.5 vs. −1.7 95%CI −2.2 to −1.3; *p* < 0.0001). Other secondary endpoints were met such as more than a 50% reduction in the mean moderate to severe migraine days in the last month of follow-up and better scores in the migraine specific quality of life questionnaire. The drug had a good tolerability with nausea and nasopharyngitis being slightly more common in the active group. Elevation of hepatic enzymes occurred in four patients taking the active drug and two taking the placebo; an increase of alanine aminotransferase more than 3-fold the normal upper limit.

No safety concerns have been raised following the completion of the clinical trials. In line with this, study 201 involved 1798 patients that had participated in studies 301 and 302 continuing with rimegepant as an acute medication for a longer period that ranged between 6 and 12 months. Patients could take rimegepant 75 mg daily if necessary. Nevertheless, participants experienced an average of a minimum of two attacks a month, but there is not enough data on intake on more than 15 days per month, which is also the studied dosage for migraine prevention [23,25,38]. The long-term assessment of quality of life, migraine disability and productivity was positive in patients who used rimegepant as a their acute medication [39] which may be attributed to its preventive effect. Whether the daily intake of rimegepant can lead to further benefits as a preventive or lead to undocumented adverse effects has not yet been studied. Hepatotoxicity has not been detected in any of the trials to date and no cardiovascular issues have been reported [17,18]. In line with this, blood pressure was not affected when comparing the subcutaneous sumatriptan 6 mg combined with rimegepant 75 mg against sumatriptan alone [26].

### 3.2. Ubrogepant

Ubrogepant was approved by the FDA for the acute treatment of migraine in December 2019 [40]. Similarly, to rimegepant 75 mg, peak plasma concentrations of ubrogepant are reached within 1.5 h after an oral dose of 50 mg or 100 mg [41] which could be delayed after a meal with a high fat content [42]. The in vitro plasma protein binding is 87% [42]. It has a hepatic metabolism through the CYP3A4. In contrast to the first generation gepants, ubrogepant has two pharmacologically inert metabolites consisting of glucuronide conjugates that have a 6000-fold less potency [42]. The elimination half-life ranges from 5 to 7 h and the elimination route is essentially biliary/faecal [42]. Ubrogepant crosses the hemato-encephalic barrier with a cerebrospinal fluid plasma ratio of 0.03 and has a low CGRP receptor occupancy in monkey studies [41].

Regarding pharmacodynamics, ubrogepant has a high potency of inhibition for the human CGRP receptor, with a mean Ki (inhibitor constant, the smaller the Ki, the greater the binding affinity) of 0.067 ± 0.04 nM in SK-N-MC cells. In addition, it shows high selectivity for human CGRP receptors in comparison with receptors from other molecules of the calcitonin family, such as adrenomedullin [41]. Similar to olcegepant [43], ubrogepant also has moderate affinity and antagonist activity at the AMY1 receptor [41]. The specific sites of action of ubrogepant are not known [20]. When using dermal vasodilation response to capsaicin (CIDV), ubrogepant was able to inhibit this response in a dose-dependent manner, independent of the capsaicin concentration, with a E_max_ (maximum effect expected from the drug) for an inhibition of 0.732 (±0.0859) and an EC_50_ of 3.19 [41].

The first results of the efficacy and tolerability of ubrogepant were published in 2016 in a phase II trial in which 834 participants were randomized to treat one of their migraine attacks with ubrogepant 1 mg, 10 mg, 25 mg, 50 mg, 100 mg, or a placebo. Of these, only the 100-mg dose showed statistically significant efficacy (25.5% vs. 8.9%; *p* < 0.001) in the co-primary endpoints, namely, pain freedom, whereas headache response at two hours (severity reduction from 2–3/3 to 0–1/3) was not significant, thus, testing other doses was not applicable in the hierarchical analysis. The 25 mg and 50 mg were significant at an unadjusted *p*-value. Tolerability was similar in the treatment and placebo arms, with only mild adverse events [19]. The phase 3 studies assessing the acute treatment of one single attack were published in 2019 [20,21]. ACHIEVE I is a randomized placebo-controlled trial in which the treatment options were reduced to the two highest doses tried in the phase-II studies. The analysis included 1436 adults with low-frequency episodic migraine with or without aura. Participants could also treat a single migraine attack, but they were given the option to repeat the dose. Co-primary efficacy endpoints were also stablished at 2 h. These were headache freedom at 2 h as well as absence of the MBS out of photophobia, phonophobia, or nausea. Pain freedom was met for the 100-mg dose, 50-mg dose, and the placebo at 21.2%, 19.2%, and 11.8%, respectively, with statistically significant differences between both treatment arms and the placebo (*p* = 0.002). Freedom from the MBS was, in the same order, 37.7%, 38.6%, and 27.8% (*p* = 0.002) [20]. The ACHIEVE II trial had a similar design to ACHIEVE I and assessed 1355 participants taking a placebo, 50 mg or 25 mg of ubrogepant. In this case, both doses were statistically significant as compared to the placebo, but only the 50-mg dose was effective for the MBS as well [21]. In contrast with sumatriptan [44], participants taking a second dose of ubrogepant may have an additional benefit. If pain was still moderate after 2 h of the first dose, participants of both studies were allowed to take a second blinded dose. Among these, the data for the 50-mg arm were pooled, and showed higher pain free rates than the placebo after 2 h, with a similar rate of side effects [45]. Additionally, previous response to triptans did not interfere the efficacy of ubrogepant, although a subgroup of participants with contraindications was included in the group with insufficient response, and triptan-naïve patients had a higher placebo effect [46]. The duration of the effect was investigated in another study. Differences between ubrogepant 50 mg and the placebo were found after 1 h for pain relief, 1.5 h for absence of the MBS, and 2 h for pain freedom, and maximal differences were found after 4 h for pain relief and MBS, and after 8 h for pain freedom [47]. No differences were documented in the trials between migraine with or without aura. No other predictors of efficacy have been reported, although in a recent study assessing real-world experience, the presence of aura or episodic migraine, a low number of previous failed preventives and positive responses to treatment with onabotulinumtoxinA or monoclonal antibodies targeting the CGRP pathways could predict a good response. However, concomitant treatment with the latter may increase possible adverse events [48]. The potential role of gepants in sensitization has been recently investigated in a preclinical model of medication overuse headache evoked by sumatriptan, which was administered over more than 10 days. Indeed, ubrogepant reversed allodynia in sensitized rats in a dose-dependent manner, and when administered repeatedly, did not produce sensitization [49].

Ubrogepant was designed as a more potent drug in comparison to its predecessors, telcagepant and MK-3207 [19,50,51,52]. A phase I double-blind, placebo-controlled trial assessed the hepatic safety of ubrogepant for 8 weeks [50]. In this trial, 516 healthy participants were randomized to receive 100 mg of ubrogepant or a placebo over two days, followed by a subsequent 48 h without treatment. Adverse events were similar in both groups, and elevation of hepatic enzymes was observed in both groups. Those cases probably related to the medication were asymptomatic and resolved after continued dosing.

Ubrogepant was generally well tolerated in both phase III studies, with a higher incidence of adverse events with higher doses [20]. The most frequently reported adverse events within 48 h were nausea, dry mouth, and somnolence, these were higher on the 100-mg dose. Serious adverse events within 30 days of the medication were only reported on the treatment arms (3 in the 50-mg and 2 in the 100-mg group) [20]. Reassuringly, a 52-week open label scheme that followed both phase III trials in which participants were allowed to treat a maximum of 8 attacks per month showed no hepatotoxicity, and some participants described headache, oropharyngeal pain, and upper respiratory infections [53]. Interactions should be considered with CYP3A4 inhibitors and inducers. Co-administration with a strong CYP3A4 inhibitor such as ketoconazole or a moderate inhibitor such as verapamil caused an increase in ubrogepant. CYP3A4 inducers such as rifampicin resulted in a reduction in ubrogepant exposure [42]. Ubrogepant should be avoided in pregnant patients. Doses should be adjusted in patients with severe renal impairment [42].

Gepants share the mechanism of action with antibodies targeting the CGRP pathway, and the possibility of their combination and eventual interaction has been a matter of discussion. Ubrogepant has recently been shown to be safe and tolerated when combined with antibodies targeting the CGRP receptor and ligand, such as galcanezumab and erenumab, respectively. In this study, 40 patients were randomized to take a single dose of ubrogepant 100 mg or a placebo a week before the injection with the antibody, and 4 days later, three consecutive daily doses. AUC and side effects were similar [54]. Indeed, ubrogepant may be the acute migraine medication that has the least amount of drug-drug interactions [55].

### 3.3. Atogepant

Atogepant was the first gepant developed exclusively as a preventive treatment for migraine targeting the CGRP pathway.

Atogepant is rapidly absorbed, with a median T_max_ of 1–2 h [56] and its C_max_ can increase in patients with hepatic impairment [56]. Plasma protein binding varies slightly depending on the hepatic function, with a range from 95.3% in patients with severe impairment to 98.2% in healthy patients [56]. Its elimination half-life is ~11 h, which is comparable to that of rimegepant [28].

Its affinity at the CGRP receptor in humans is higher than that of ubrogepant, with a lower Ki: 15–26 pM. In line with the abovementioned gepants, atogepant also has an affinity at the AMY1 receptor, although it is 100-fold less than at the CGRP receptor [24].

The first phase IIb/III trial was completed in 2018 and consisted of a randomized, multicenter, double-blind, placebo-controlled, parallel-group study [24]. The study included 795 participants with episodic migraine that were randomized to six groups consisting of the combination of a dose of placebo and one daily dose of atogepant 10 mg, 30 mg or 60 mg, two daily doses of atogepant 30 mg or 60 mg, or two doses of a placebo, over 12 weeks [24]. The primary endpoint was the change from baseline in mean monthly migraine days (MMD), which was met for the five doses. At the time of writing, several abstracts are available as first results of the new phase III trials. The phase III trial ADVANCE (NCT03777059) [57], analysed 873 participants with episodic migraine, randomized in this occasion to four groups, consisting of atogepant at 10 mg, 30 mg or 60 mg doses or a placebo. The efficacy endpoint was similar to the phase IIb/III trial and was fulfilled in all the branches. A secondary endpoint of ≥50% reduction in MMD was achieved by 56%, 59%, 61%, and 29%, respectively, reaching statistical significance (*p* < 0.001) in all groups against the placebo. Data on quality of life were also extracted from this cohort by the means of questionnaires, with a reduction in emotional impact and daily functioning [58,59]. The combination with onabotulinumtoxinA has shown to be effective in reducing sensitization and cortical spreading depression in an animal model [60]. Results from other phase III trials on episodic migraine (NCT03700320) and chronic migraine (NCT03855137) are still pending.

Atogepant was overall safe and well tolerated. The frequency of side effects reported was 18–54% of patients in the active branches, versus 16–57% in the placebo groups. The main side effects included nausea and fatigue, which were dose dependent, and constipation [24,57]. Long-term use of atogepant 60 mg for one year was tolerated, with 18% of adverse events considered to be related to the drug, namely upper respiratory tract infection, constipation, nausea, and urinary tract infection. None of the serious adverse events were considered related to the drug [61]. When administered in a single oral dose of 60 mg, atogepant was well tolerated in patients with hepatic impairment, with similar plasma levels, but increased systemic exposures to atogepant did not have a clinical translation [56]. The repeated administration of supra-therapeutic doses of 170 mg of atogepant during 28 days was well tolerated, with no serious adverse events attributable to the drug and no elevation of hepatic enzymes [62]. Atogepant seems to have a low profile of pharmacological interactions. At a 60-mg dose there is no interaction with sumatriptan 100 mg, but it could delay the T_max_ and the C_max_ may be lower [63]. The drug may be administered with contraceptive medication. This may increase the AUC of levonorgestrel, which would not have a clinical impact [64]. Pharmacokinetic information for rimegepant, ubrogepant, and atogepant is summarised in Table 2.

### 3.4. Zavegepant

Zavegepant (BHV3500-201), formerly vazegepant [13], is the first third generation gepant. Its Ki for CGRP is 0.023 nM. Because of the structure of zavegepant, different routes of administration are under study, and these include subcutaneous, oral, and intranasal. The latter has recently undergone a phase IIb/III RCT (NCT03872453) of which the initial results were positive. The intranasal formulation was developed looking for a rapid onset of action [13]. Zavegepant 10 mg and 20 mg were found efficacious and reached the primary endpoints for efficacy (see Table 1); the 10-mg dose led to pain freedom at 2 h in 22.5% of the patients as compared to 15.5% with the placebo (*p* = 0.0113) and the 20-mg was, similarly, superior to the placebo (23.1% vs. 15.5%; *p* = 0.0055). Likewise, freedom of the MBS at 2 h was achieved by 41.9% and 42.5% of patients on 10 mg, 20 mg and the placebo, respectively (*p* < 0.05). The fast onset of action was proven, and pain relief was documented as early as 15 min after intake with sustained efficacy at 2 h. There is no published data on whether the efficacy is sustained after 48 h, which would be in line with previous gepants. No safety concerns have been reported. Dysgeusia was the most typical side effect reported in 13.5%, 16.1%, and 3.5% of patients in the 10-mg, 20-mg, and placebo arms, respectively. In line with the other new generation gepants, there were no hepatic issues [22]. Because of the poor oral bioavailability of zavegepant, a series of azepinone-based compounds have been tested, with one of them showing similar affinity for CGRP receptors and improved oral bioavailability in rats (F_PO_ = 17% vs. 1.7% for zavegepant) [65].

## 4. Discussion

The presented evidence shows the efficacy of rimegepant, ubrogepant, and zavegepant as acute migraine medications as well as atogepant and rimegepant as migraine preventives. Previous reviews have also addressed the drugs’ characteristics and the available evidence to support their use [13,66]. From a pharmacokinetic/pharmacodynamics perspective, a comparison of rimegepant, ubrogepant, and atogepant can be seen in Table 2. These drugs have comparable data with the exception of the elimination half-life, which is similar for rimegepant and atogepant (~11 h) both of which can have a preventive indication. The question on whether these new generation gepants will substitute triptans in the future has not been addressed as no clinical trials have been designed to show non-inferiority of gepants vs. triptans. Nevertheless, the only RCT including 100 mg sumatriptan against different doses of rimegepant suggested slightly more efficacy of the former [35]. The utility of gepants as migraine preventives is in line with the efficacy of blocking the CGRP pathway using monoclonal antibodies [67]. Indeed, the use of an acute medication that can also act as a migraine preventive, such as rimegepant, seems particularly appealing in patients with medication overuse headache where the traditional approach was based on acute medication withdrawal which was not always successful [68]. Data on safety and tolerability derived from the clinical trials point to safe and well tolerated drugs. Future pharmacovigilance studies may confirm these data.

At this stage, we can say that the use of gepants as acute treatment is a reasonable alternative for migraineurs that are not responsive to triptans and for those with a contraindication for triptans use such as cardiovascular diseases [69].

## Figures and Tables

**Table 1 jcm-11-01656-t001:** Summary of clinical trials on gepants.

**Acute Medications**	**Study (Reference)**	**Phase**	**Pain Freedom at 2 h (%)**	**Absence of MBS at 2 h (%)**	**Total % of AEs**
Rimegepant	[15]	II	75 mg: 31.5 150 mg: 32.9 300 mg: 29.7 Placebo: 15.3	* 52.3 * 44.7 * 51.4 * 28.1	7 6 5 7
	Study 301 [16]	III	75 mg: 19.2 Placebo: 14	36.6 27.7	12.6 10.7
	Study 302 [17]	III	75 mg: 19.6 Placebo: 12	37.6 25.4	17.1 14.2
	Study 303 [18]	III	75 mg: 21 Placebo: 11	35 27	13.2 10.5
Ubrogepant	[19]	IIb	100 mg: 25.5 Placebo: 8.9	60.8 42	24.5 20.4
	ACHIEVE I [20]	III	50 mg: 19.2 100 mg: 21.2 Placebo: 11.8	38.6 37.7 27.8	9.4 16.3 12.8
	ACHIEVE II [21]	III	25 mg: 20.7 50 mg: 21.8 Placebo: 14.3	34.1 38.9 27.4	9.2 12.9 10.2
Zavegepant	[22]	II/III	10 mg: 22.5 20 mg: 23.1 Placebo: 15.5	41.9 42.5 33.7	13.5 16.1 3.5
**Preventive Medications**	**Study (Reference)**	**Phase**	**Change in MPM (Days)**		**Total % of AEs**
Rimegepant	[23]	III	75 mg every other day: −4.3 Placebo: −3.5		36 36
Atogepant	[24]	IIb/III	10 mg QD: −4 30 mg QD: −3.76 60 mg QD: −3.55 30 mg BD: −4.23 60 mg BD: −4.14 Placebo: −2.85		18 21 23 21 26 16

MBS: most bothersome symptom. AEs: adverse events. MMD: mean monthly migraine/probable migraine days. * MBS was not utilized in this study, and we present data on phonophobia at 2 h. See the text.

**Table 2 jcm-11-01656-t002:** Summary of pharmacokinetics of gepants.

Gepant	T_max_	Plasma-Protein Binding	Metabolism	Elimination Half-Life	Excretion
Rimegepant [15,17,25,26,27,28]	1.5 h	96%	Hepatic (CYP3A4)	11 h	Faeces
Ubrogepant [41,42]	1.5 h	87%	Hepatic (CYP3A4)	5–7 h	Faeces
Atogepant [24,56]	1–2 h	98.2%	Hepatic (CYP3A4)	11 h	Faeces > urine *

T_max_: Time to maximum concentration. * When specifically tested with a radioactive tracer, 81% was fecal and 8% urinary.

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
