# Peer review of "New Generation Gepants: Migraine Acute and Preventive Medications"

_jcm, 2022, doi:10.3390/jcm11061656_

Round 1

Reviewer 1 Report

Authors kindly answered all the queries previously submitted.

Reviewer 2 Report

Thank you for sufficiently addressing the items in question in the manuscript.

This manuscript is a resubmission of an earlier submission. The following is a list of the peer review reports and author responses from that submission.

Round 1

Reviewer 1 Report

The authors present a timely and succinct update on the evolving field of gepants in migraine management. The senior author is a/the leading world expert in this particular area as well as multiple others with respect to migraine pathophysiology and management, thus the readership will benefit greatly from his evidence based insights into this new generation of migraine treatment. 

The brief reviews of pharmacokinetics/dynamics of these medications is a useful and unique contribution in this article.  

A few minor suggestions: 

  1. The title should reflect the nature of the review - would the authors clarify if this is a narrative or scoping review - either in the title or in the methods section of the abstract and paper. 
  2. The abstract has at least two spelling errors (the PDF does not allow me to reference which lines) "The" and "development" 
  3. In Methods section please clarify the timeline of the search period for this review "from inception" is not a clear date. Which years were included in the search (xxxx-2021). 
  4. The Discussion section ends somewhat abruptly it may benefit from a concluding sentence summarizing the intent of the review and the overall thoughts of the authors re: gepants in migraine in the future. 
  • Please do another close spell check and grammatical review of the manuscript (including abstract), there were some spelling errors and minor English grammatical errors that require revision. 
  • The review may benefit from one further table briefly comparing the four gepants discussed with respect to pharmacokinetics/dynamics in a "quick look" format as much detail has been presented within the text of the review but readers may prefer a snapshot view of the actual chemical and functional differences between these gepants in tabular form.  
  • The authors may wish to reference previous recent reviews on gepants in migraine (Negro & Martelletti 2019, Moreno-Ajona et al. 2020, Medicine in Drug Discovery)

Reviewer 2 Report

The review summarised phase I-III trials on next-generation gepants. The authors brilliantly exposed pharmacokinetic data and clinical outcomes of each gepant.

The review has been conducted rigorously, and results are presented in detail.

I suggest few minor revisions to the manuscript.

Included trials evaluated rimegepant, ubrogepant and zavegepant. Please specify in the introduction or methods section that studies evaluating first-generation gepants (olcegepant, telcagepant, and other compounds) were excluded.

Please add some details (at least number of participants) for rimegepant study 301.
Please report statistical significance, if available, for studies on atogepant.

Discussion should be expanded. A brief comparison of the presented gepants could help the reader summarise the results (i.e., factors that, at least theoretically, indicate the advantage of one gepant on the others, such as pharmacokinetic or interactions). Please consider adding a sentence about the safety of next-generation gepants in the discussion.

Minor suggestions: please correct the following typing errors in the abstract
"Th calcitonin-gene-related peptide..." with "THE..."
"development" with "development"